# Antagonistic control of intracellular signals by EpOMEs in hemocytes induced by PGE$_2$ and their chemical modification for a potent insecticide

**Niayesh Shahmohammadi[1], Shiva Haraji[1], Falguni Khan[1], Åshild Moi Sørskår[2], Parastoo Ebrahimi Danielsen[2], Anders Vik[2], Yonggyun Kim[1]\***

**1** Department of Plant Medicals, Andong National University, andong, Republic of Korea, **2** Department of Pharmacy, Section for Pharmaceutical Chemistry, University of Oslo, Oslo, Norway

\* hosanna@anu.ac.kr

## Abstract

During an infection, prostaglandin E$_2$ (PGE$_2$) mediates immune responses in insects and later epoxyoctadecamonoenoic acids (EpOMEs) are produced from linoleic acid to suppress excessive and unnecessary immune responses. Intracellular signaling pathway by which these oxylipins suppress the immune responses was previously unclear. This study demonstrated that EpOMEs antagonize the secondary messengers induced by PGE$_2$ in a lepidopteran species, *Maruca vitrata*. PGE$_2$ injections significantly increased hemocyte-spreading behavior, along with raised calcium ion and cAMP levels in hemocytes, and also up-regulated phenoloxidase activity and expressions of antimicrobial peptides. These cellular and humoral immune responses induced by PGE$_2$ were dose-dependently inhibited by EpOMEs, with 12,13-EpOME being more effective than 9,10-EpOME in immunosuppression. PGE$_2$ treatment also elevated the total number of circulating hemocytes, with the majority (88.4%) of these increased hemocytes being granulocytes. Conversely, EpOMEs suppressed the up-regulation of total hemocyte count induced by PGE$_2$ and directly reduced the total hemocyte count by inducing apoptosis in granulocytes, as visualized by the TUNEL assay. These immunosuppressive and cytotoxic effects suggest the potential of EpOME as a lead compound for developing a novel type of insecticides. To chemically stabilize EpOMEs, the epoxide group was replaced with a propoxide group, and the carboxylic terminal was methylated. The 12-propoxyl regioisomer was selected based on immunosuppressive bioassays. Further investigation of the two possible enantiomers of 12-propoxyl regioisomer showed that the 12*R*-enantiomer was more effective than the 12*S*-enantiomer in immunosuppression. The resulting 12*R*-propoxy octadecamonoenoic methyl ester displayed insecticidal activities at low nanogram levels per insect by hemocoelic injection and at < 50 ppm by the leaf-dipping method against three lepidopteran insects.

**Data availability statement:** All relevant data are within the paper and its Supporting Information files.

**Funding:** This work was supported by a grant (No. 2022R1A2B5B03001792) from the National Research Foundation (NRF) funded by the Ministry of Science, ICT and Future Planning, Republic of Korea. This study was also funded by a research grant from Andong National University.

**Competing interests:** The authors have declared that no competing interests exist.

## Introduction

Insect immunity is innate, and its programmed recognition receptors activate specific infection signals upon exposure to various pathogens [1]. These infection signals are then relayed to immune-associated tissues such as hemocytes and the fat body, eliciting cellular and humoral immune responses [2]. Locally and systemically, these signals utilize NO, biogenic monoamines, cytokines, or oxylipins to mediate the response [3]. Notably, oxylipins play a pivotal role in integrating and specifically enhancing these infection signals [4].

Oxylipins, a family of oxygenated polyunsaturated fatty acids, include eicosanoids and epoxyoctadecenoic acids (EpOMEs), which are originated from arachidonic acid and linoleic acid, respectively [5,6]. Eicosanoids are subdivided into prostaglandins (PGs) oxygenated by cyclooxygenase, leukotrienes oxygenated by lipoxygenase, and epoxyeicosatrienoic acids oxygenated by epoxygenase. Particularly, PGs are extensively studied for their roles in mediating various insect physiological processes, including immune responses [7]. A specific receptor for $PGE_2$ was first identified in the lepidopteran insect, *Manduca sexta* [8], and its deletion mutant exhibited significant immunosuppression during the larval stage and a lack of oogenesis in female adults of *Spodoptera exigua* [9]. Ahmed and Kim [10] investigated the signal transduction in hemocytes following $PGE_2$ exposure, where cAMP and calcium ion levels were elevated, leading to actin cytoskeleton rearrangement and increased water in-current through aquaporin. The trimeric G proteins associated with the $PGE_2$ receptor have been delineated in lepidopteran insects [11].

EpOMEs are generated through the catalytic activity of cytochrome P450 monooxygenase (CYP), which yields two regioisomers of coronaric acid (9,10-EpOME) and vernolic acid (12,13-EpOME), known respectively as leukotoxin and isoleukotoxin [12]. These EpOMEs are subsequently hydrolyzed by soluble epoxide hydrolase (sEH), resulting in the formation of dihydroxy-octadecamonoenoates (DiHOMEs) [13,14]. Both EpOMEs and DiHOMEs have been identified in *S. exigua*, along with genes associated with the enzymes responsible for their synthesis and degradation [15,16]. Subsequent detections of EpOMEs occurred in another lepidopteran, *Maruca vitrata*, and a thrips species, *Frankliniella occidentalis* [17,18]. These findings suggest that EpOMEs play vital roles in regulating the insect physiological processes. Notably, EpOMEs are produced during the late infection stage and modulate excessive and unnecessary immune responses [15,19]. Inhibition of *sEH* using RNA interference (RNAi) or specific inhibitors resulted in fatal inflammation in insects [19] suggesting that EpOMEs function similarly to vertebrate resolvins. Intriguingly, the tomato spotted wilt virus exploits insect EpOMEs to dampen the anti-viral responses of its vector, *F. occidentalis*, facilitating its proliferation and transmission within the host [18]. These observations related to the immunosuppressive properties of EpOMEs imply an antagonistic functional relationship between EpOMEs and $PGE_2$. Yet, the mechanism by which EpOMEs negatively affect $PGE_2$'s action in target cells remains unresolved.

In addition to the immunosuppressive properties of EpOMEs, they also exhibit cytotoxic effects on insect cells [17]. Although EpOMEs displayed modest insecticidal activity, alkoxide derivatives including methoxy, ethoxy, propoxy, and iso-propoxy ethers demonstrated significant potency and enhanced the efficacy of other microbial insecticides such as *Bacillus thuringiensis* and baculovirus [16]. Among these derivatives, the propoxy methyl ester was the most potent immunosuppressor and showed relatively high cytotoxic effects. This selected derivative comprised a mixture of four regioisomers. Synthesis and evaluation of individual 12- and 13-propoxy regioisomers would be valuable for future activity assessments. Additionally, exploring individual enantiomers of the most promising regioisomer could reveal any preferential activity towards one enantiomer over the other.

This study explored the antagonistic activity of EpOMEs against $PGE_2$ in signal transduction within target cells. Given the immunosuppressive and cytotoxic properties of EpOMEs, we evaluated propoxy derivatives to identify the lead compound for developing a novel insecticide.

## Materials and methods

### Insect rearing

The larvae of three lepidopteran species were used in this study: *M. vitrata*, *S. exigua*, and *Plutella xylostella*. The larvae of *M. vitrata* and *S. exigua* were reared on artificial diets [20,21] under laboratory conditions, $25 \pm 2°C$ and $60 \pm 5\%$ relative humidity, with a photoperiod of 16:8 h (L:D), undergoing five instars (L1-L5). The larvae of *P. xylostella* were reared on a cabbage diet under the same conditions, undergoing four instars (L1-L4). All adults received 10% sucrose for oviposition.

### Chemicals

Three hormones: 9,10-EpOME, 12,13-EpOME, and prostaglandin $E_2$ ($PGE_2$) were procured from Cayman (Ann Arbor, MI, USA). Bovine serum albumin (BSA), dimethyl-sulfoxide (DMSO), and t-octylphenoxy-polyethoxyethanol (Triton X-100) were sourced from Sigma Aldrich Korea (Seoul, Korea). An anticoagulant buffer (ACB, pH 4.5) was prepared using 186 mM NaCl, 17 mM Na 2 EDTA, and 41 mM citric acid. Phosphate-buffered saline (PBS) was formulated with 100 mM phosphoric acid, adjusted to pH 7.4 with 1 N NaOH. 5-bromo-2'-deoxyuridine (BrdU) and anti-BrdU antibody were acquired from Abcam (Shanghai, China). Terminal deoxynucleotidyl transferase (TdT) and FITC-conjugated anti-mouse IgG antibody were purchased from Thermo Fisher Scientific (Wilmington, DE, USA). 4,6-diamine-2-phenylindole dihydrochloride (DAPI), cAMP analog (8-(4-chlorophenylthio) adenosine 3,5′-cyclic monophosphate sodium salt), and calcium ionophore salt (A23187) were obtained from Sigma-Aldrich Korea. Fura-8AM was sourced from AAT Bio quest (Sunnyvale, CA, USA) and dissolved in 100% DMSO.

### Chemical synthesis of EpOME-mimics

A regio-isomer mixture of EpOME analog (A841) was synthesized as reported in a previous study [16]. The processes for synthesizing PD23, PD28, AS45, and AS56 are briefly discussed below and illustrated in Fig 1. Detailed spectroscopic data and chromatographic methods for these chemical syntheses are provided in S1 Document. To synthesize PD23 (Fig 1A), Wittig salt **2** was prepared through the esterification of commercially available 9-bromononanoic acid (**1**). The resultant methyl ester was converted into Wittig salt **2** using triphenylphosphine in refluxing acetonitrile. Hexanal (**3**) was then reacted with allyl magnesium bromide, yielding the secondary alcohol **4**. This alcohol was transformed into the propoxy ether **5** via a Williamson ether synthesis employing 1-propylbromide. Hydroboration-oxidation of the alkene in **5** produced the primary alcohol **6**, which was subsequently oxidized to aldehyde **7** using Dess-Martin Periodinane. PD23 was finally synthesized through a Wittig reaction between aldehyde **7** and the ylide from Wittig salt **2**.

PD28 was synthesized as depicted in Fig 1B. Addition of heptanal (**8**) to a solution of vinylmagnesium bromide yielded the secondary alcohol **9**. A Williamson ether synthesis on **9** with 1-propylbromide produced the ether **10**. A hydroboration-oxidation on the alkene in **10** produced the primary alcohol **11**, which was oxidized with Dess-Martin Periodinane to yield

**(A)**

**(B)**

**(C)**

**Fig 1. Schematic pathways for the synthesis of EpOME alkoxide regioisomers.** (A) The diagram illustrates the synthesis process for Wittig salt 2 and PD23. (B) The diagram details the synthesis of PD28. (C) The diagram outlines the synthesis of AS56.

aldehyde **12**. A Wittig reaction between aldehyde **12** and the ylide of Wittig salt **2** produced PD28.

The *R*-enantiomer (AS56) of ethyl (*Z*)-12-propoxyoctadec-9-enoate was synthesized by asymmetric synthesis using (-)-Ipc$_2$B(allyl)borane in a Brown-allylation on heptanal (**8**), as depicted in Fig 1C. This produced (*R*)-**4**. The secondary alcohol (*R*)-**4** was converted into the ether (*R*)-**5** using a Williamson ether synthesis. Subsequently, epoxidation of the terminal alkene in (*R*)-**5** with *meta*-chloroperoxybenzoic acid (*m*CPBA) yielded the epoxide **13** as a mixture of two diastereomers. Cleavage of the epoxide in **13** with periodic acid produced aldehyde (*R*)-**12**. Finally, a Wittig reaction between aldehyde (*R*)-**12** and the ylide of **2** produced the desired *R*-enantiomer AS56. The synthesis of its enantiomer, AS46, followed a similar process using (+)-Ipc$_2$B(allyl)borane for the Brown-allylation step.

## Quantification of cAMP and Ca²⁺ signals

cAMP measurement was conducted using a cyclic AMP ELISA kit (Cayman). Hemolymph (50 μL) from *M. vitrata* larvae was combined with 148 μL of ACB, to which 2 μL of PGE$_2$ (10$^{-7}$ M) was added. One hour later, the mixture was initially centrifuged at 1,000 × *g* for 10 min at 4°C. The cell pellet was lysed in 100 μL of 0.1 M HCl for 20 min at room temperature (RT), and the supernatant was collected following centrifugation at 5,000 × *g* for 10 min at 4°C. Subsequently, 50 μL of 0.4 M KOH and acetic anhydride from the kit were used for acetylation of the supernatant. Following acetylation, 50 μL of each sample or standard was dispensed into the designated wells, followed by the sequential addition of 50 μL of cAMP AChE Tracer and 50 μL of cAMP EIA antiserum. The plate was incubated overnight at 4°C. After five washes with 200 μL of washing buffer, 200 μL of Ellman's reagent was added and the samples were incubated until color development occurred under darkness at RT. Absorbance was measured at 405 nm. cAMP concentration was calculated by using a regression equation derived from known standard concentrations. The unknown concentration was subsequently calculated using this equation based on the value observed at 405 nm.

To study Ca$^{2+}$ signaling in hemocytes of *M. vitrata* in response to various EpOME regio-isomers, hemolymph was treated with 2 μL of Fura-8AM (1 mM) in addition to each treatment condition (1 μg/mL). An hour post-treatment, the hemolymph was mounted on a slide glass using 2.5% paraformaldehyde. Fura-positive cells were visualized using a fluorescence microscope (DM2500, Leica, Wetzlar, Germany) at 200×magnification. Fluorescence intensity was quantified using ImageJ software (https://imagej.nih.gov/ij), with each measurement replicated three times.

### Terminal deoxynucleotidyl transferase dUTP nick-end labeling (TUNEL) assay

A TUNEL assay to determine apoptosis was conducted using an *in situ* Cell Death Detection kit (Abcam, Cambridge, UK). For this assay, 1 μg of test chemicals was injected into L5 larvae and incubated for 24 h. After the incubation period, hemolymph was collected into ACB as previously described. Subsequently, 10 μL of hemocyte suspension was placed on a cover glass in a wet chamber and mixed with 1 μL of 5-bromouridine (BrdU) (10 μM) solution containing terminal transferase. The mixture was incubated at RT for 1 h, followed by a medium replacement with 4% paraformaldehyde and a further 10-min incubation at RT. The cells were then washed with PBS and treated with 0.3% Triton-X in PBS, followed by a 2-min incubation at RT. After blocking with 5% BSA in PBS for 10 min, the cells were incubated with mouse anti-BrdU antibody (diluted 1:15 in blocking solution) for 1 h at RT. The unbound anti-BrdU antibody was removed, and the cells were then incubated with FITC-conjugated anti-mouse IgG antibody (diluted 1:500 in blocking solution) for 1 h. Following another PBS wash, 10 μL of DAPI was added and the mixture was incubated at RT for 5 min. The cells were washed again with 10 μL of PBS. Subsequently, a glycerol: PBS (1:1) solution (10 μL) was added to the cells on a cover glass, which was then placed on a slide glass for observation under a fluorescence microscope (DM2500, Leica) in FITC mode. Each treatment was replicated three times.

### RNA extraction, cDNA preparation, and qPCR

To avoid contamination from non-target organisms, the intestine was removed from the larvae before RNA extraction. Total RNA was extracted from the larvae using TRIzol reagent (Invitrogen, Carlsbad, CA, USA) following the manufacturer's instructions. This RNA was then used to synthesize complementary DNA (cDNA) using an RT-premix (Intron Biotechnology, Seoul, Korea) that included an oligo-dT primer. The cDNA was quantified with a spectrophotometer (Nano Drop, Thermo Fisher Scientific, Wilmington, DE, USA). A quantity of 100 ng of cDNA was used as a template for quantitative PCR (qPCR) with gene- specific primers. The qPCR was performed using a SYBR Green real-time PCR master mix (Toyobo, Osaka, Japan), following the protocol described by Bustin et al. [22] on a Step One Plus Real-Time PCR System (Applied Biosystems, Singapore). The qPCR protocol started with a heat treatment at 95°C for 10 min denaturation, followed by 40 cycles at 95°C for 30 s, an annealing step at 52°C for 30 s, and an extension at 72°C for 30 s. The actin gene served as an endogenous control. Each treatment was replicated three times with independent samples. Expression analysis via qPCR utilized the comparative CT method [23].

### Total hemocyte counts

L5 larvae of *M. vitrata*, which had been injected with a test compound at a dosage of 1 μg per larva. Four hours after the injection, hemolymph was collected from the larvae using a capillary, which involved puncturing the prolegs. The hemolymph samples were subsequently loaded onto a hemocytometer, and hemocyte counts were conducted using a phase-contrast

microscope (CKX31, Olympus, Tokyo, Japan) at 400× magnification. This procedure was replicated three times for each treatment.

### Hemocyte-spreading behavior analysis

Total hemolymph (250 μL) was extracted from *M. vitrata* larva and collected into 750 μL of ACB. The hemocyte suspension was chilled on ice for 30 min. Following centrifugation at 800 × *g* for 5 min, 700 μL of the supernatant was discarded, and the cell pellet was gently re-suspended in 700 μL of TC-100 insect tissue culture medium (Welgene, Gyeongsan, Korea). A 9 μL sample of the hemocyte suspension was combined with $PGE_2$ ($10^{-7}$ M) and various EpOME regioisomers (1 μg/mL) on a glass coverslip and then incubated in a wet chamber in darkness. The cells were fixed with 4% paraformaldehyde for 10 min at RT. After three PBS washes, the cells were permeabilized with 0.2% Triton X-100 in PBS for 2 min at RT, followed by a PBS wash and a subsequent block with 10% BSA in PBS for 10 min at RT. After another PBS wash, the cells were incubated with FITC-tagged phalloidin for 1 h at RT. Following three more PBS washes, the cells were incubated with DAPI (1 mg/ mL) in PBS for nuclear stain-ing. After two final washes in PBS, the cells were observed under a fluorescence microscope (DM2500, Leica) at 200× magnification. Hemocyte-spreading was assessed by measuring the extension of F-actin beyond the original cell boundary. Every behavior assay involved 100 ran-domly selected cells. Each treatment was replicated three times, based on three measurements with independently prepared samples.

### Nodulation assay

*M. vitrata* larvae were used in bioassays to determine hemocyte nodule formation. *Escherichia coli* Top10 was injected at a dose of $2 \times 10^4$ cells/larva through the proleg and incubated at RT for 8 h. The insects were also injected with test compounds (1 μg/larva) along with *E. coli* cells to assess effects on nodulation. Following the dissection of the test larvae, melanized nodules were counted using a stereoscopic microscope (Stemi SV11, Zeiss, Jena, Germany) at 50× magnification. Each treatment was conducted in triplicate.

### Phenoloxidase (PO) activity

PO activity was measured using L-3,4-dihydroxyphenylalanine (DOPA) as a substrate. Each L5 larva of *M. vitrata* was inoculated with *E. coli* ($2 \times 10^4$ cells/larva). 8 h later, hemolymph was extracted from the treated larvae, and the plasma was isolated by centrifugation at 800 × *g* for 5 min at 4°C. The reaction mixture (200 μL) consisted of 10 μL of plasma, 4 μL of test compound (1 μg/mL), 10 μL of DOPA (1 M), and 176 μL of PBS. Absorbance was measured at 495 nm using a VICTOR multi-label Plate reader (PerkinElmer, Waltham, MA, USA). The activity was quantified as the change in absorbance per min per microliter of plasma (ABS/min/μL). Each assay was replicated three times.

### Antimicrobial peptide (AMP) gene expression analysis

The levels of four AMP genes were analyzed in L5 larvae of *M. vitrata* following an immune challenge with *E. coli* ($2 \times 10^4$ cells/larva) injected into the hemocoel. In conjunction with *E. coli*, the insects were also injected with test compound(s) to evaluate their effect on AMP induction. Total RNA was collected 8 h after treatment as previously described. Following cDNA synthesis, qPCR was conducted using the same method as above, employing primers specific to the AMP genes [17]. The actin gene was amplified as the housekeeping gene, serv-ing as the internal standard in the qPCR assay. All samples were analyzed in triplicate.

## Insecticidal analysis

Comparative toxicity analysis of different EpOME alkoxide regioisomers was conducted by injection and oral feeding in *M. vitrata*, *S. exigua*, and *P. xylostella*. Three doses (10, 100, and 1,000 ng/larva) of 12,13-EpOME, A841, PD23, PD28, AS46, and AS56 were administered to the last instar larva of the tested insects. For oral administration, various concentrations (10, 100, and 1,000 µg/mL) of these regioisomers were prepared with DMSO. Prior to feeding, larvae were starved for 5 h to ensure rapid consumption of the treated diet. Inoculation was achieved by soaking the diet in the test compound suspensions for 3 min. A control group received a soaking diet in DMSO. The treated diets were then placed on filter paper for approximately 5 min to remove excess moisture. Mortality was evaluated five days after treatment. Each treatment involved 10 larvae and was repeated three times. Subsequently, larvae were maintained at 25 ± 2°C.

## Effect of R-enantiomer (AS56) on the virulence of *Bacillus thuringiensis*

A non-lethal concentration (100 ppm) of *B. thuringiensis kurstaki* (Thuricide, Bayer Crop Science, Seoul, Korea) was introduced to L5 larvae of *S. exigua* together with AS56 (10 ppm) through oral feeding. Mortality was assessed five days after treatment (DAT). Each treatment involved 10 larvae and was replicated three times. Following the treatment, larvae were maintained at 25 ± 2°C.

## Behavioral bioassay using Y-tube olfactometry

To evaluate the behavioral responses of *S. exigua* to AS56, a Y-tube olfactometer was employed. In each experiment, 20 L3 larvae received a treatment of 1,000 ppm AS56 via leaf feeding. At 24 h post-treatment, larvae were observed in the Y-tube. Positive responses were noted when insects fed on cabbage plants. Those that did not proceed past the fork point were recorded as no response ('NR'). All trials were carried out in the dark at 25 ± 1°C and 65% RH, each lasting 20 min and replicated three times.

## Statistical analysis

cAMP and $Ca^{2+}$ signals were analyzed by one-way ANOVA. Percentage data were transformed using arcsine for normalization and subjected to ANOVA analysis. Mean comparisons were made using the least squares difference (LSD) test via PROC GLM of the SAS program [24] and discriminated at Type I error = 0.05. All graphs in this study were created using Graph-Pad Prism v. 8.0.1 (Boston, MA, USA).

## Results

### Antagonistic activity of EpOME against $PGE_2$ in immunity

An eicosanoid, $PGE_2$, mediated the hemocyte-spreading behavior of *M. vitrata* by extending cytoplasmic projections with elevated F-actin levels (Fig 2A). In contrast, 12,13-EpOME inhibited this behavior. Furthermore, the addition of 12,13-EpOME also counteracted the effects mediated by $PGE_2$. The antagonistic actions of the two EpOMEs were dose-dependent ($F = 20.0$; df = 12, 28; $P < 0.0001$), with 12,13-EpOME proving more potent ($F = 10.0$; df = 1, 28; $P = 0.0038$) than 9,10-EpOME (Fig 2B).

Calcium ion levels were elevated in hemocytes of *M. vitrata* following exposure to $PGE_2$ (Fig 3A). This $Ca^{2+}$ mobilization induced by $PGE_2$ was antagonized by EpOME. The inhibitory effect of EpOME was dose-dependent ($F = 62.4$; df = 12, 28; $P < 0.0001$), with 12,13-EpOME being more effective ($F = 33.0$; df = 1, 28; $P < 0.0001$) than 9,10-EpOME (Fig 3B). The elevated cAMP levels

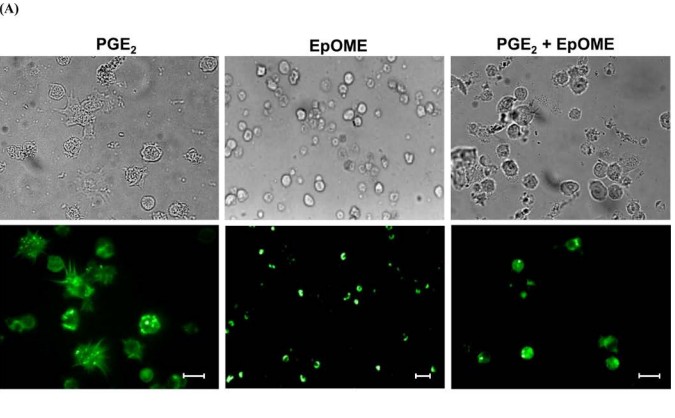

**(A)**

**(B)**

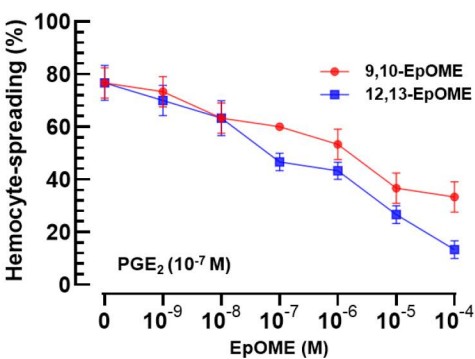

**Fig 2. Antagonistic activities of EpOMEs against PGE$_2$ modulation of hemocyte-spreading behavior in *M. vitrata*.** (A) Spread hemocytes in DIC (upper panels) and FITC (lower panels) modes under PGE2 exposure at 10-7 M and 12,13-EpOME at 10-4 M. The scale bars represent 10 μm. (B) Dose-response of EpOMEs on the modulation of PGE$_2$ at 10$^{-7}$ M to the spreading behavior. Hemocyte spreading was determined by counting cells exhibiting F-actin growth (stained by FITC) beyond the cell boundary from 100 randomly selected cells. Each treatment was replicated three times.

induced by PGE$_2$ were also diminished following exposure to EpOMEs in a dose-dependent manner ($F$ = 213.7; df = 12, 28; $P$ < 0.0001), with 12,13-EpOME proving more effective ($F$ = 68.1; df = 1, 28; $P$ < 0.0001) than 9,10-EpOME (Fig 3C). The impaired hemocyte-spreading behavior was restored with the addition of a cAMP analog or Ca$^{2+}$ ionophore (Fig 3D).

To examine the inhibitory effects of EpOME on the humoral immune responses of *M. vitrata*, PO enzyme activity and AMP expression levels were assessed (Fig 4). PO activity was also enhanced by adding PGE$_2$ to hemolymph (Fig 4A). Both EpOMEs significantly inhibited PO activation, with 12,13-EpOME being more effective than 9,10-EpOME. Four AMPs showed increased expression in *M. vitrata* following PGE$_2$ injection (Fig 4B). An addition of 12,13-EpOME suppressed this expression upsurge. Nonetheless, the decrease in AMP expression levels was reversed with the addition of a cAMP analog or Ca$^{2+}$ ionophore. Surprisingly, EpOME's immunosuppressive activity led to significant septicemia in *M. vitrata* larvae following exposure to a nonpathogenic bacterial infection with *E. coli* (Fig 4C).

## Hemolytic activity of EpOME in *M. vitrata*

In response to EpOMEs, hemocytes were analyzed in *M. vitrata* larvae (Fig 5). Naïve L5 stage larvae exhibited 1.02 × 10$^6$ hemocytes per mL of hemolymph (Fig 5A). Following a bacterial infection, the total hemocyte count (THC) doubled within 4 h. Similarly, PGE$_2$ injection increased

THC in the same timeframe. The hemocytes were categorized into four morphological types (granulocyte, plasmatocyte, oenocytoid, and spherulocyte) to assess the differential hemocyte counts (DHC) across the three treatments: naïve, immune-challenged, and PGE$_2$-induced (Fig 5B). The bacterial challenge resulted in a significant alteration in DHC compared to that of naïve larvae ($X^2 = 8.4$; df = 3; $P = 0.035$), particularly increasing the proportion of granulocytes. PGE$_2$ injection also significantly altered the DHC compared to naïve larvae ($X^2 = 8.1$; df = 3; $P = 0.045$), increasing the proportion of granulocytes. However, no significant differences were noted in the DHCs between the immune-challenged and PGE$_2$-induced larvae ($X^2 = 0.5$; df = 3; $P = 0.915$). Both EpOMEs significantly inhibited the up-regulation of THC in response to PGE$_2$, with 12,13-EpOME being more potent ($F = 5.41$; df = 1, 16; $P = 0.002$) than 9,10-EpOME (Fig 5C). Interestingly, EpOME treatment also altered the DHC ($X^2 = 8.5$; df = 3; $P = 0.037$) compared to that of PGE$_2$-induced larvae (Fig 5D). The addition of EpOMEs restored the DHC to levels similar to those of naïve larvae ($X^2 = 1.8$; df = 3; $P = 0.615$).

**(A)**

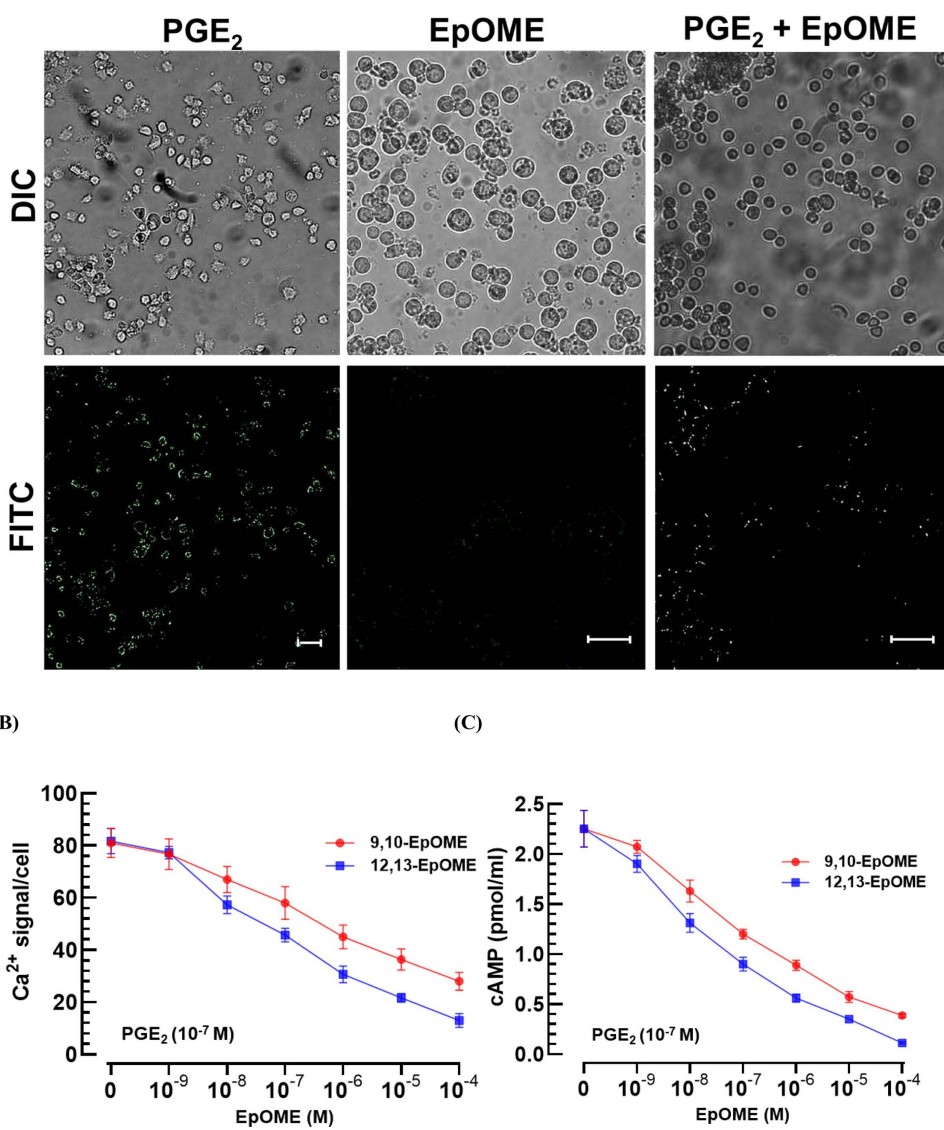

**(D)**

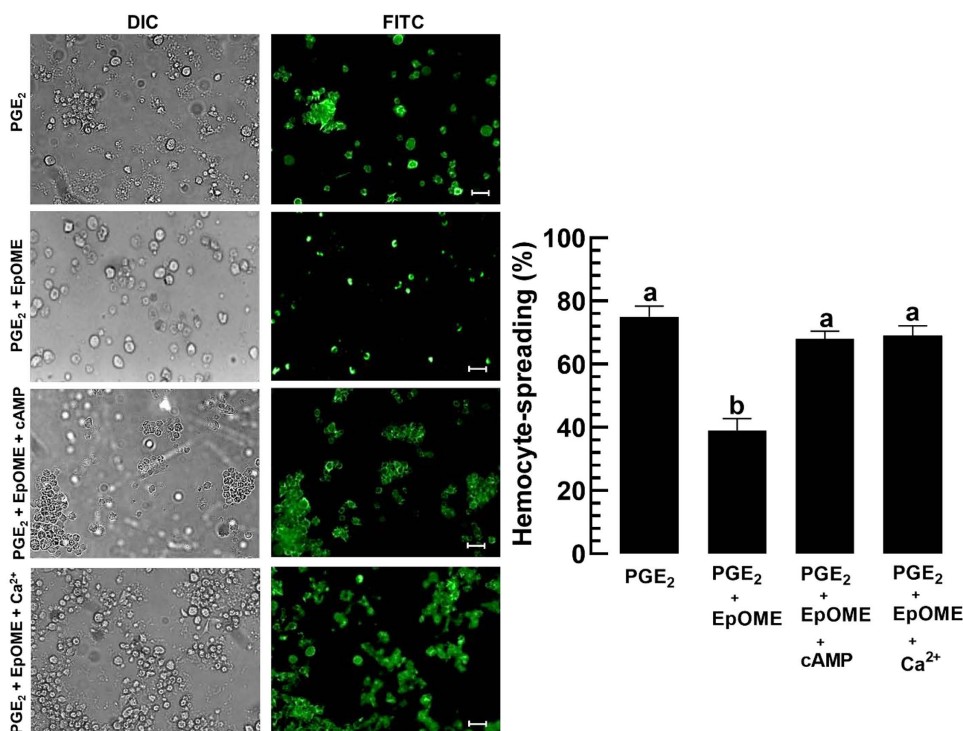

**Fig 3. Antagonistic activities of EpOMEs against PGE$_2$ modulation of intracellular secondary messengers (Ca$^{2+}$ and cAMP) in hemocytes of *M. vitrata*.** (A) Hemocytes viewed in DIC mode with Ca2+ signals in FITC mode, under the influence of PGE2 at 10-7 M and 12,13-EpOME at 10-4 M. Scale bars represent 10 μm. (B) Dose responses of EpOMEs to Ca2+ levels in hemocytes, under a constant concentration of PGE2, where Ca2+ signals were measured by normalized FITC intensities against DAPI intensities. (C) Dose responses of EpOMEs to cAMP levels in hemocytes, maintained at a steady concentration of PGE2, where cAMP levels were adjusted for cell density. Each experimental condition was replicated three times. (D) Rescue effects of cAMP analog or calcium ionophore against EpOME-induced suppression in hemocyte-spreading behavior. Treatments involved PGE2 at 10-7 M and 12,13-EpOME at 10-4 M. The cAMP analog used was 8-(4-chlorophenylthio) adenosine 3,5′-cyclic monophosphate at 1 mM, and the calcium ionophore was A23187 at 1 mM. Hemocyte-spreading behavior was evaluated by counting cells showing F-actin extensions beyond (stained by FITC) the cell boundary in 100 randomly selected cells. Each treatment was replicated three times. Distinct letters above the standard error bars designate significant differences among means at a Type I error of 0.05 (LSD test). Scale bars represent 10 μm.

EpOMEs demonstrated hemolytic activity in *M. vitrata* (Fig 6). THC decreased in a dose-dependent manner ($F = 70.3$; df = 6, 16; $P < 0.0001$) with EpOME injection (Fig 6A). To investigate the cytotoxic effects of EpOMEs on hemocytes, apoptosis was analyzed using the TUNEL assay (Fig 6B). Both EpOMEs induced apoptosis, as evidenced by DNA fragmentation visualized by FITC fluorescence in the TUNEL assay. Among the two EpOMEs, 12,13-EpOME was more effective than 9,10-EpOME at inducing apoptosis (Fig 6C). Most (> 80%) of the apoptotic cells were granulocytes (Fig 6D).

## Variation in the immunosuppressive activity of EpOME alkoxide isomers

To obtain stable EpOME analogs, the unstable epoxide was substituted with an alkoxide, and the terminal carboxyl group was methylated [16]. Various alkoxide groups were assessed, including methoxy, ethoxy, propoxy, and iso-propoxy. Each alkoxide was synthesized into four regioisomers, with an alkoxide group positioned at the 9th, 10th, 12th, or 13th positions. From these isomeric

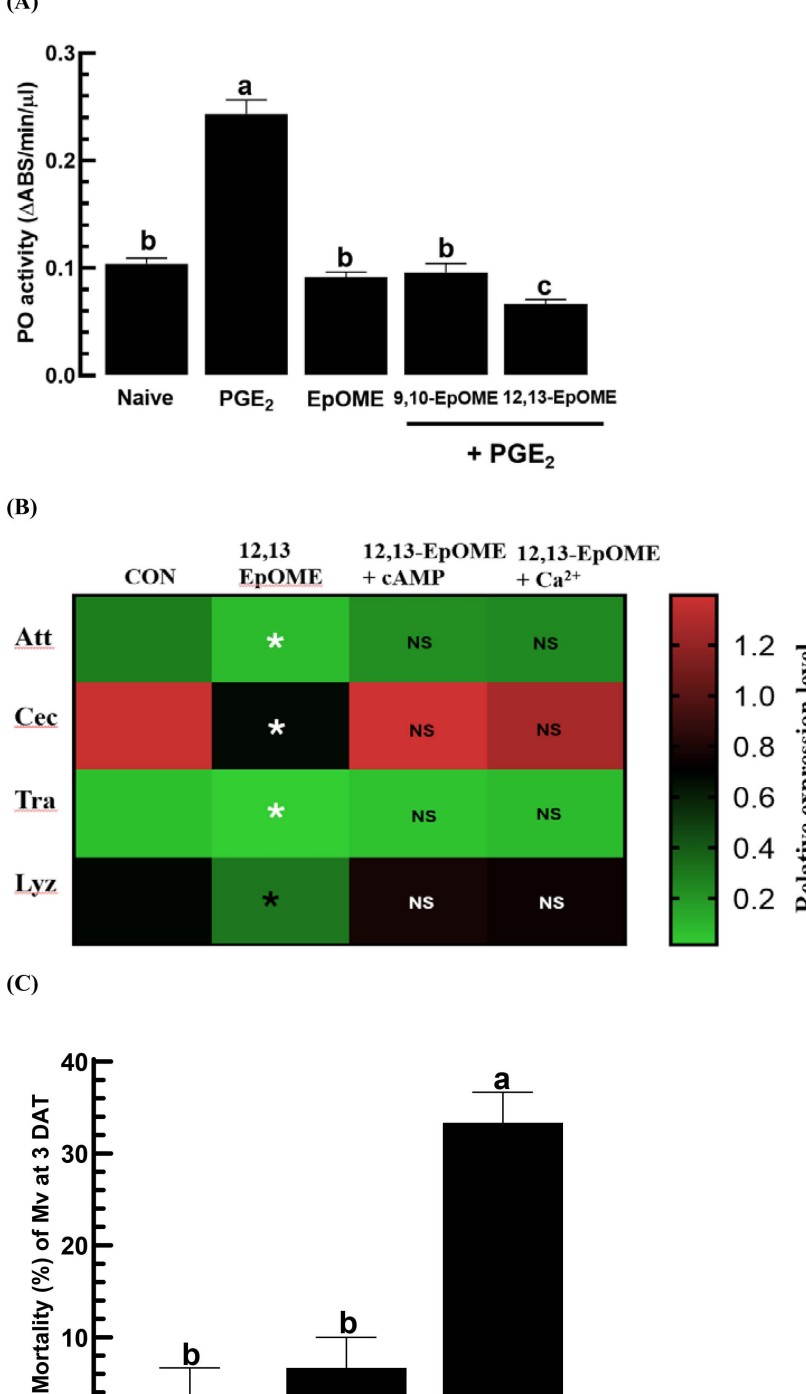

**(A)**

**(B)**

**(C)**

**Fig 4. Antagonistic activities of EpOMEs against the modulation of PGE₂ on the humoral immune responses of _M. vitrata._** (A) Their antagonistic effects on phenoloxidase (PO) activity through the addition of EpOME (1 μg/larva) to an injection of PGE2 (1 μg/larva). PO activity in the plasma was measured 4 h post-injection. Each treatment was replicated three times. (B) Suppressive effect of 12,13-EpOME on the expression of antimicrobial peptide ('AMP') genes: attacin ('Att'), cecropin ('Cec'), transferrin ('Tra'), and lysozyme ('Lyz'). AMP expression was assessed 8 h

after immune challenge through injecting E. coli (2 × 104 cells/larva) together with 12,13-EpOME, cAMP analog (8-(4- chlorophenylthio) adenosine 3,5′-cyclic monophosphate), and calcium ionophore (A23187). Asterisks indicate significant differences compared to the control ('CON') at a Type I error of 0.05 (LSD test). (C) Septicemia induced by nonpathogenic *E. coli* (2 × $10^4$ CFU/larva) with addition of EpOMEs (1 μg/larva). Mortality was measured at 3 days post-treatment. Each treatment was replicated three times with 10 larvae per replication. Distinct letters above the standard error bars delineate significant differences among means at a Type I error of 0.05 (LSD test).

mixtures, A841 (Fig 7A) was chosen for its high efficacy in immunosuppressing *S. exigua* [16]. A841 effectively suppressed calcium ion (Fig 7B) and cAMP levels (Fig 7C) induced by $PGE_2$ in the hemocytes of *M. vitrata*. This study also compared the two pure regio-isomers 12-propoxy ('PD28') and 13-propoxy ('PD23') and demonstrated that PD28 outperformed PD23 in suppressing two intracellular signal molecules. PD28, a chiral molecule, exists as a mixture of two enantiomers (mirror images). Consequently, each enantiomer of PD28 was synthesized through asymmetric organic synthesis, yielding the 12*S*-enantiomer (AS46) and the 12*R*-enantiomer (AS56). Of these, AS56 outperformed AS46 in suppressing two intracellular signal molecules.

The comparative analysis of the isomers was conducted using cellular and humoral immune assays (Fig 8). In experiments on hemocyte-spreading behavior, EpOME alkoxides displayed more potent inhibitory activities than natural 12,13-EpOME, presenting dose-dependent inhibition where PD28 and AS56 were the most effective (Fig 8A). Similarly, PD28 and AS56 showed the strongest inhibitory activities, with AS56 significantly outperforming PD28 ($F$ = 22.74; df = 1, 16; $P$ = 0.0002) (Fig 8B). In cellular immune responses, PD28, PD23, AS46, and AS56 demonstrated greater inhibitory activities than 12,13-EpOME or A841 (Fig 8C). In humoral immune responses, all EpOME alkoxides effectively inhibited the AMP induction in response to bacterial challenges (Fig 8D). These results indicated that all EpOME alkoxides were more effective than 12,13-EpOME in suppressing immune responses, with AS56 being one of the most potent inhibitors.

## Variation in the hemolytic activity of EpOME alkoxide isomers

All EpOME alkoxides exhibited greater activity than 12,13-EpOME in the cytotoxic activity assay of *M. vitrata* hemocytes (Fig 9A). Among the alkoxides, AS56 showed the most potent hemolytic activity. From a TUNEL assay, the enhanced cytotoxicity of the alkoxides was attributed to their induction of hemocyte apoptosis, with AS56 being the most effective (Fig 9B).

## Insecticidal activity of a specific EpOME alkoxide isomer, AS56

Insecticidal activities of the EpOME alkoxides were analyzed using three lepidopteran insects. By injecting these compounds into the larvae, all EpOME alkoxides demonstrated significantly higher insecticidal activities than 12,13-EpOME in these insects (Fig 10). PD28 and AS56 proved to be the most potent, with $LD_{50}$ values ranging from 5 to 19 ng/larva against *P. xylostella* and *M. vitrata* five days post-treatment (S1 Table). *S. exigua* exhibited higher tolerance, where PD28 and AS56 showed insecticidal activity at approximately 178 and 10 ng/larva, respectively. These EpOME alkoxides also exhibited high insecticidal activity when applied to insects via the leaf-dipping method (S2 Table). AS56 was also highly potent at $LC_{50}$ values around 50 ppm against all three species five days after treatment.

## Toxic activity of AS56 on larval feeding behavior

Oral treatment with AS56 demonstrated increased susceptibility in the L1 and L3 larval stages of *S. exigua* compared to the L5 stage (Fig 11A). In the Y-tube assay, larvae treated with AS56 displayed reduced appetite, showing a significant non-preference for a diet treated with AS56

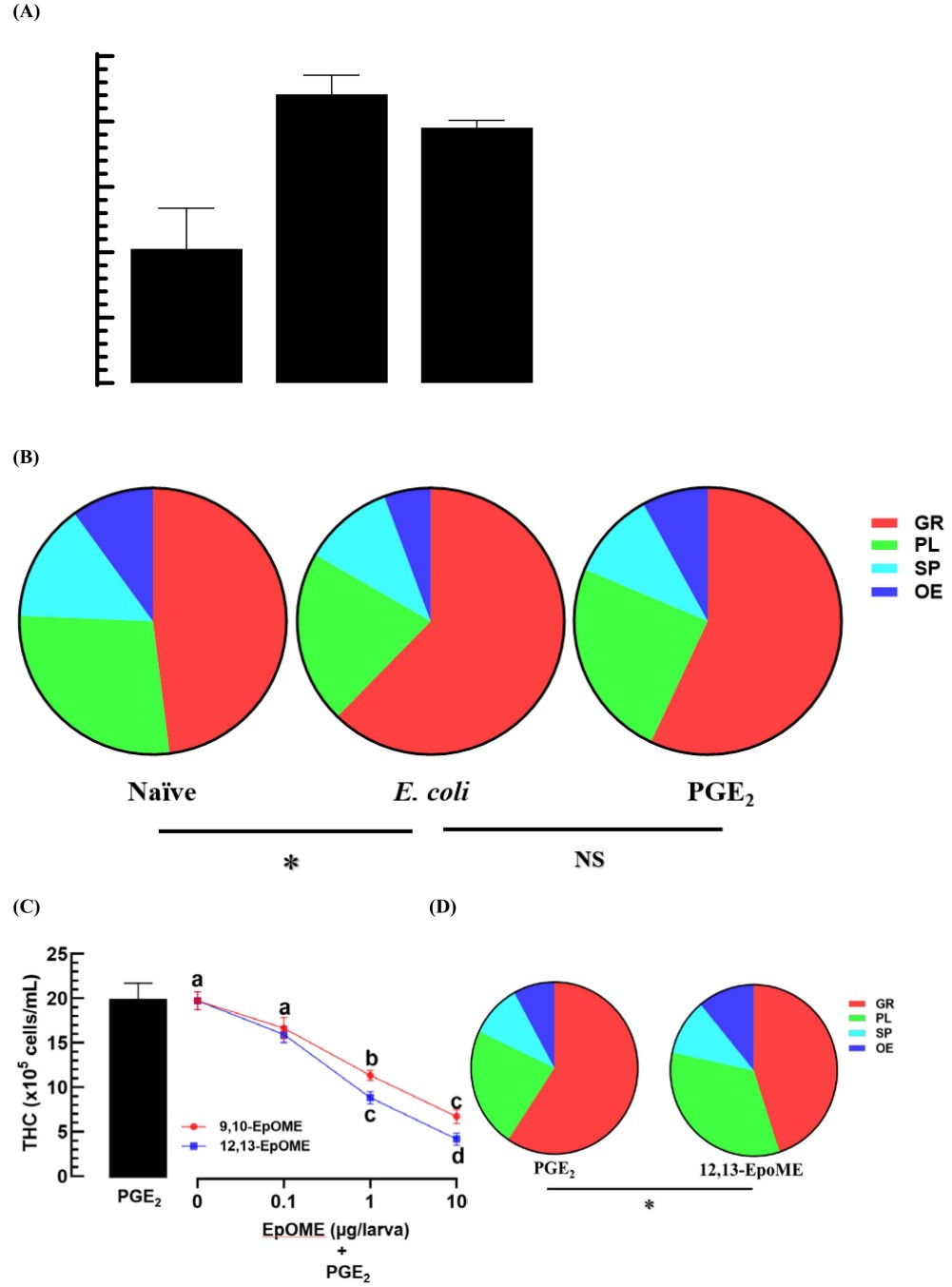

**Fig 5. Cytotoxic activities of EpOMEs on the hemocytes of *M. vitrata*.** (A) This panel shows the up-regulation of total hemocyte counts ('THC') following injections of *E. coli* ($2 \times 10^4$ CFU/larva) or PGE$_2$ (1 μg/larva). (B) This section depicts their effects on differential hemocyte count ('DHC'): granulocyte ('GR'), plasmatocyte ('PL'), spherulocyte ('SP'), and oenocytoid ('OE'). (C) This part demonstrates the dose-response of EpOMEs on THC and DHC. Distinct letters above the standard error bars or asterisks signify significant differences among the means at a Type I error rate of 0.05 (LSD test). 'NS' denotes no significant difference.

([Fig 11B]). It was noteworthy to observe a high rate of no response in the choice test with the AS56-treated diet. This reduction in feeding behavior was also apparent in the feeding area analysis, where untreated L3 larvae consumed almost twice as much cabbage as those

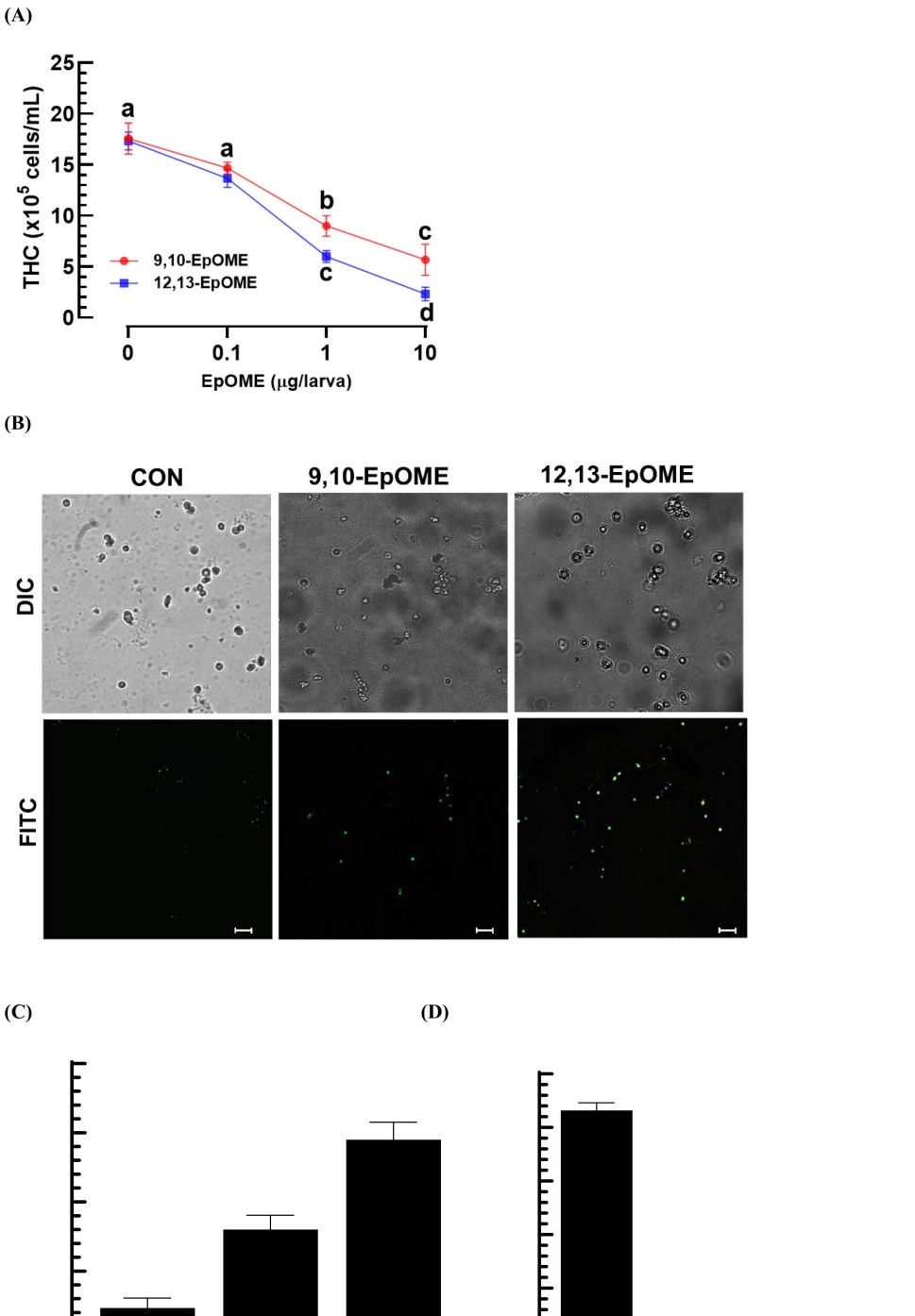

**Fig 6. Hemolytic activity of EpOMEs against *M. vitrata* via apoptosis.** (A) Cytotoxicity of EpOMEs on total hemocyte counts ('THC'). (B) TUNEL assays of hemocytes treated with EpOME (1 μg/larva) are shown here, with the control ('CON') using DMSO solvent. BrdU is detected by FITC, while DIC reveals the hemocytes. The scale bars indicate a length of 10 μm. (C) Quantified apoptosis in response to EpOME, where a positive TUNEL signal was quantified based on FITC intensities normalized by DAPI intensities. ImageJ software (https://imagej.nih.gov/ij/) was employed to estimate signal intensities across various images. Each treatment was replicated three times. (D) Quantified apoptotic cells in hemocyte populations are presented, including: granulocyte ('GR'), plasmatocyte ('PL'), spherulocyte ('SP'), and oenocytoid ('OE'). Different letters near the standard error bars signify significant differences among the means at a Type I error rate of 0.05 (LSD test).

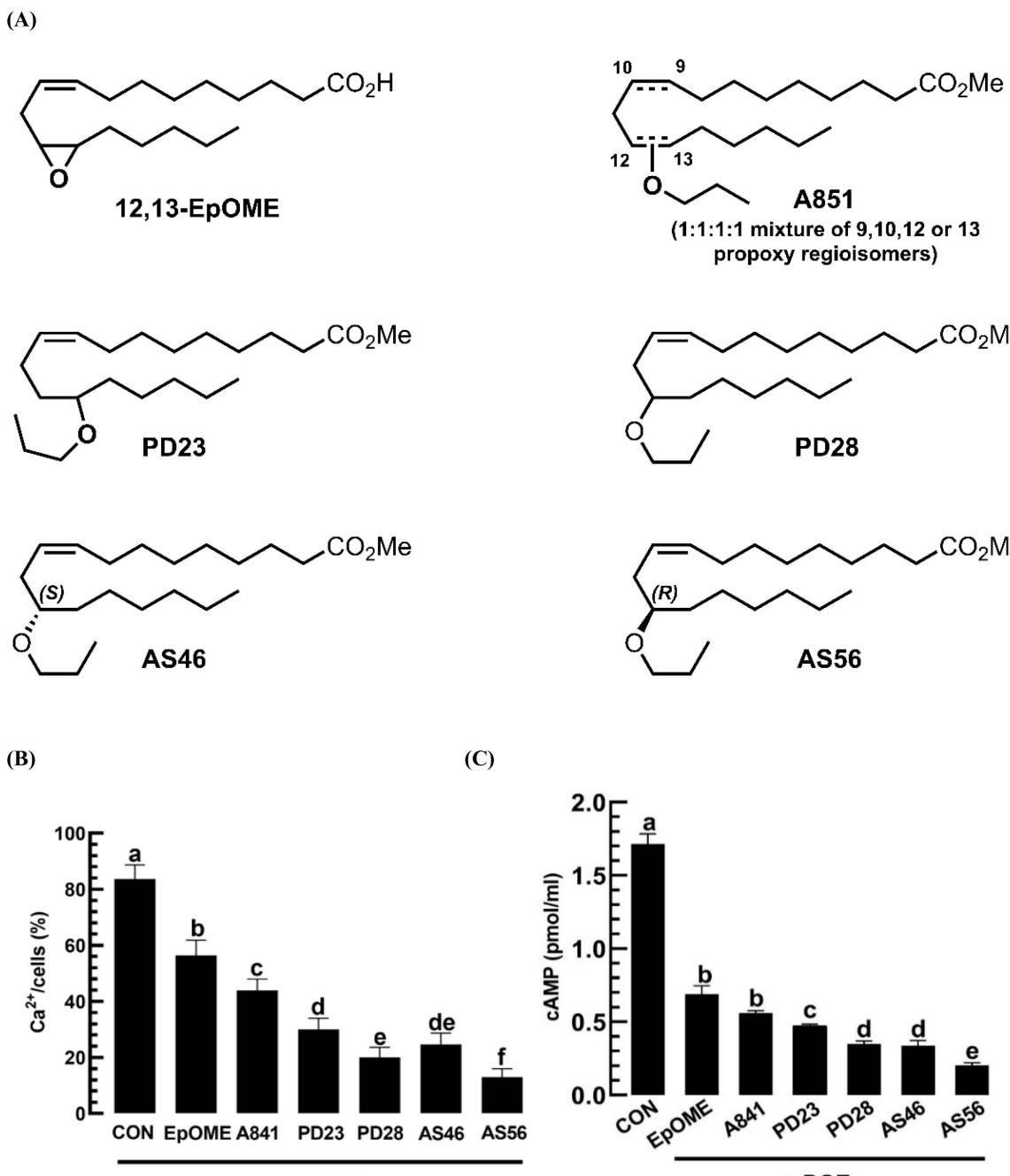

**Fig 7. Inhibitory activities of EpOME derivatives against PGE$_2$ mediation of intracellular secondary messengers in hemocytes of *M. vitrata*.** (A) This panel displays the chemical structures of EpOME and its analogs: A841 (a mixture of four propoxy analogs of EpOME), PD23 (13-propoxy), PD28 (12-propoxy), AS46 (12S-propoxy octadecamonoenoic methyl ester), and AS56 (12R-propoxy octadecamonoenoic methyl ester). (B) The inhibition of Ca2 + level in hemocytes, induced by PGE2 (10-7 M), by the addition of an EpOME analog (1 µg/mL) is shown here. The Ca2 + signal was quantified using FITC intensities normalized by DAPI intensities. The control ('CON') utilized DMSO solvent. (C) This panel demonstrates the inhibition of cAMP levels in hemocytes, also induced by PGE2 (10-7 M), through the addition of an EpOME analog (1 µg/mL). A mixture (2 µL) consisting of PGE2 (10-7 M) and each of the test compounds (1 µg/mL) was added to the hemocytes, where the cAMP level was quantified at 405 nm. Each treatment was replicated three times. Various letters above the standard error bars indicate significant differences among the means at a Type I error of 0.05 (LSD test). The scale bars measure 10 µm.

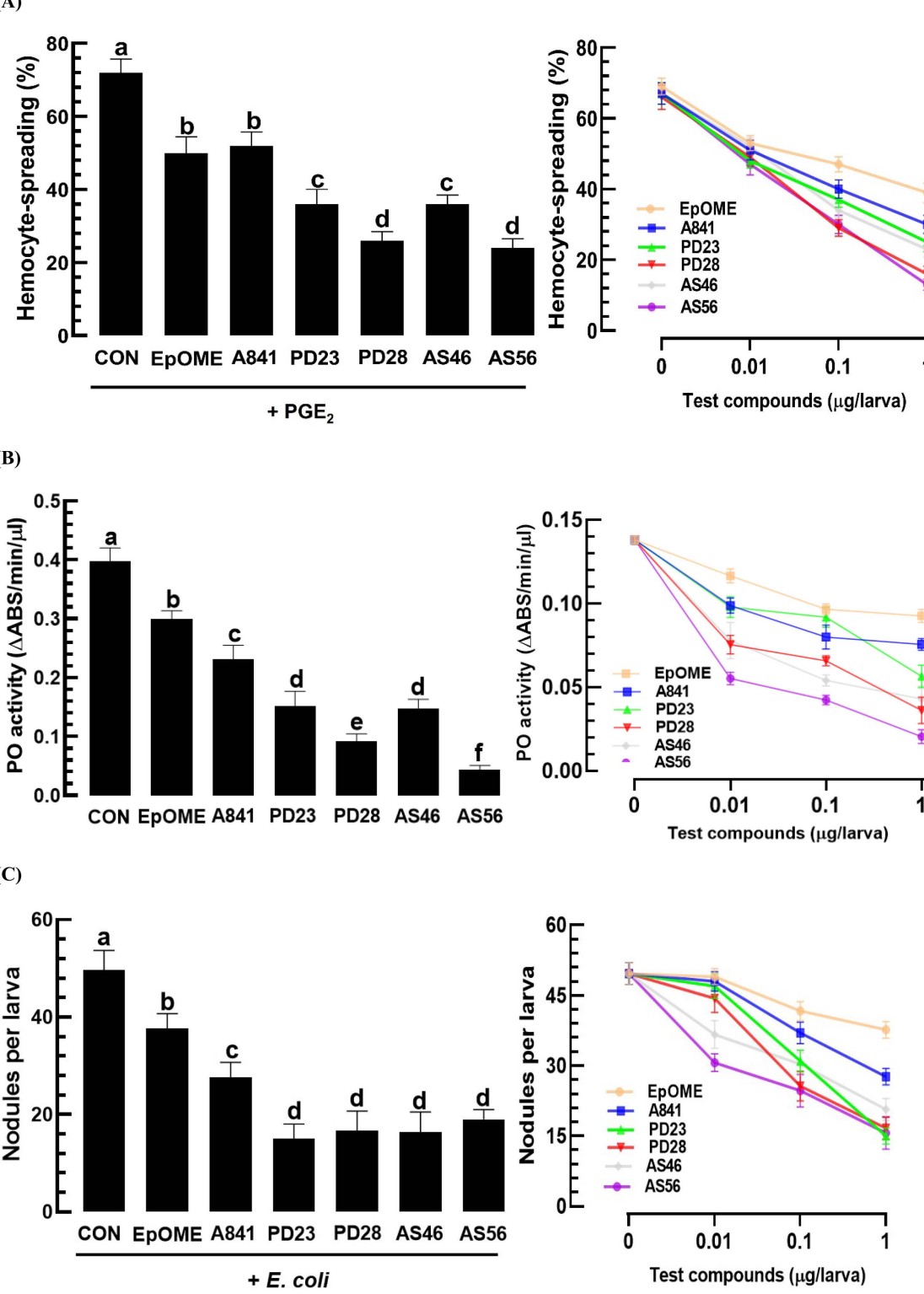

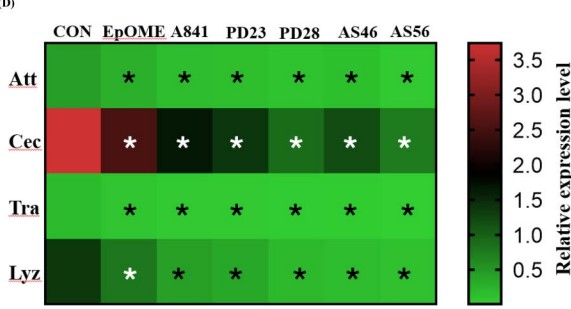

**Fig 8. Variation in immunosuppressive activities of EpOME derivatives against *M. vitrata*.** (A) Antagonistic effect of EpOME alkoxide regioisomers (1 μg/mL) (left panel) and their dose responses (right panel) on the modulation of PGE2 at 10-7 M influencing behavior. Cellular spread was determined by counting cells showing F- actin expansion (stained with FITC) outside the cell boundary among 100 randomly selected cells. (B) Antagonistic effect of EpOME alkoxide regioisomers (1 μg/larva) (left panel) and their dose responses (right panel) on modulating bacterial infection ($2 \times 10^4$ CFU/larva) regarding nodule formation. (C) Antagonistic effect of EpOME alkoxide regioisomers (1 μg/larva) (left panel) and their dose responses (right panel) on the modulation of bacterial infection ($2 \times 10^4$ CFU/larva) concerning phenoloxidase ('PO') activity. (D) Antagonistic effect on expression of antimicrobial peptide ('AMP') genes: attacin ('Att'), cecropin ('Cec'), transferrin ('Tra'), and lysozyme ('Lyz'). AMP expression was evaluated 8 h post-immune challenge by injecting *E. coli* ($2 \times 10^4$ cells/larva) along with EpOME alkoxide regioisomers at 1 μg/larva. Asterisks indicate significant differences compared to the control ('CON') at Type I error = 0.05 (LSD test). Each treatment was replicated three times.

treated with AS56 (Fig 11C). The effects were consistent in the AS56-treated welsh onion, which exhibited protection from larval feeding damage. Moreover, adding AS56 significantly increased the virulence of the microbial pesticide *B. thuringiensis*, against L5 larvae of *S. exigua* (Fig 11D, S3 Table).

## Discussion

EpOMEs act like vertebrate resolvins in insects to prevent excessive immune reactions. Indeed, the addition of 12,13-EpOME to a baculovirus significantly augmented viral infectivity against diverse lepidopteran insects [18]. Moreover, inhibiting the degradation of EpOME increased the bacterial virulence of *Bacillus thuringiensis* against lepidopteran insect pests [17]. This insight led to the hypothesis that a stable compound among EpOME derivatives could be developed as a novel insecticide. This study involved modifying natural EpOMEs into chemically stable alkoxide derivatives and screening the potent derivative to support this hypothesis.

EpOMEs exhibit immunosuppressive activity in lepidopteran insects by antagonizing immune responses mediated by $PGE_2$, inhibiting secondary messenger levels in target cells. $PGE_2$ binds to its receptor on the hemocyte membrane in the nanomolar range and upregulates cAMP/$Ca^{2+}$ levels in lepidopteran insects [7]. These secondary messengers facilitate cell shape changes for hemocyte-spreading behavior. $Ca^{2+}$ orchestrates actin cytoskeleton rearrangement in the cytoplasm, while cAMP activates aquaporin at the cytoplasmic membrane, increasing local cell volume of the hemocytes [10]. Treatments with 9,10-EpOME and 12,13-EpOME suppress secondary messenger levels induced by $PGE_2$, inhibiting cellular immune responses. They also reduce humoral immune responses as evidenced by decreased PO activity and AMP expressions triggered by $PGE_2$. This supports the physiological role of EpOMEs in moderating excessive immune responses in insects, similar to vertebrate resolvins during late infection stages.

EpOMEs display cytotoxic activity against insect hemocytes via apoptosis, which was supported by the TUNEL assay. An immune challenge increased the total hemocyte count in *M. vitrata*. $PGE_2$ alone also upregulated the total hemocyte number. This increase may not

**(A)**

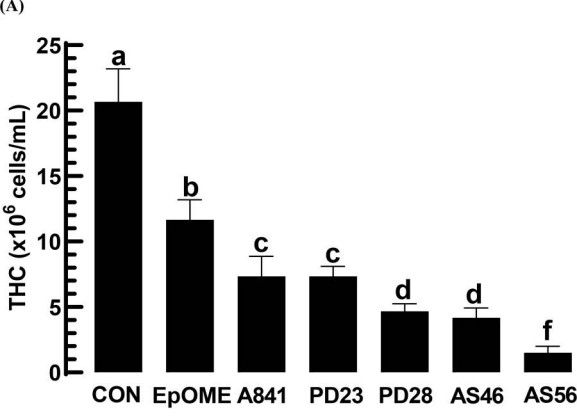

**(B)**

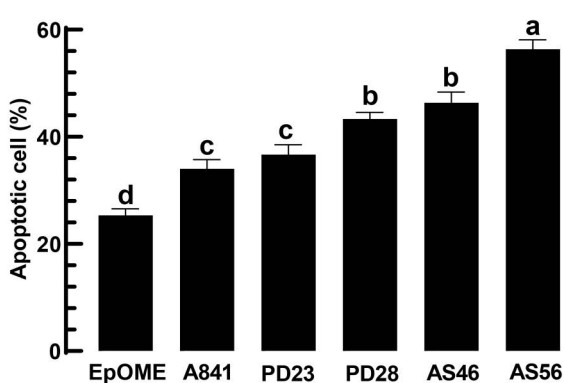

**Fig 9. Variation in hemolytic activities of EpOME derivatives against *M. vitrata*.** (A) Effect of EpOME derivatives (1 µg/larva) on total hemocyte counts ('THC') with DMSO used as the control ('CON'). (B) Cytotoxic activities of EpOME derivatives (1 µg/larva) were assessed on apoptosis by TUNEL assay. Each treatment was replicated three times. Different letters above standard error bars indicate significant differences among means at Type I error = 0.05 (LSD test).

result from *de novo* cell division via mitosis during the 4-h period, as mitosis typically requires about 24 h in most eukaryotic cells. Instead, the rise in hemocyte number likely originated from stationary hemocytes transitioning to circulatory form. Kim and Kim [25] discovered that hemocytes in a lepidopteran insect, *S. exigua*, are divided into stationary and circulatory forms. Park and Kim [26] demonstrated that $PGE_2$ facilitates the transition from stationary to circulatory form in *S. exigua*. These suggest that EpOME reduces the total hemocyte count by preventing the recruitment of the stationary hemocytes to circulatory form by the antagonistic action to the $PGE_2$ signal. Interestingly, the majority of increased hemocytes were granulocytes in immune-challenged or $PGE_2$-treated larvae. EpOME treatment reduced the total hemocyte count by inducing apoptosis, particularly targeting granulocytes. At 1 µg EpOME treatment, the total hemocyte count resembled that of naïve larvae, and differential hemocyte count was similarly consistent with naïve larvae. These findings suggest that EpOMEs play a pivotal role in maintaining the hemocyte populations in insects by preventing excessive recruitment and direct cytotoxic activity at late infection stage.

The immunosuppressive and cytotoxic activities of EpOMEs were enhanced by chemical derivatization with methylation at the carboxylic group and replacing the alkoxide at the

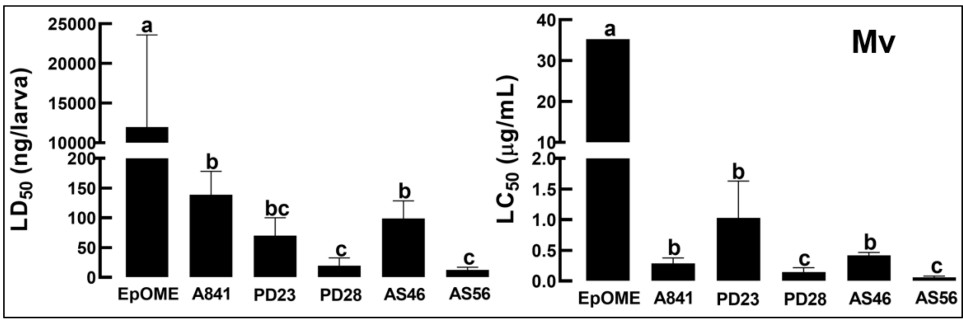

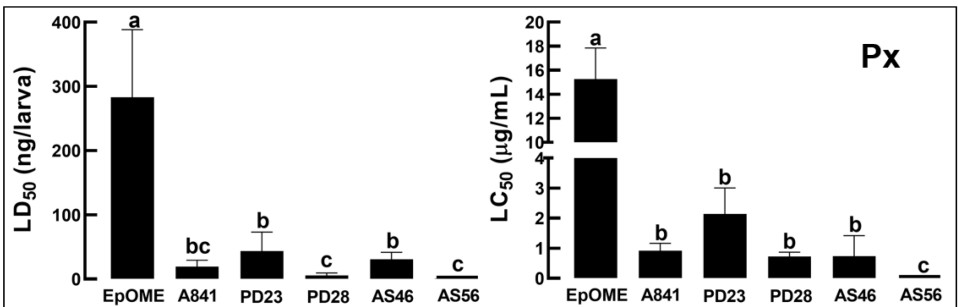

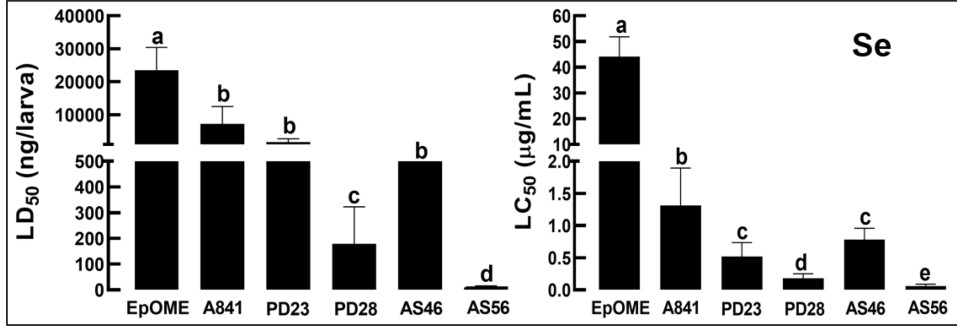

**Fig 10. Insecticidal activities of EpOME derivatives against three different lepidopteran species:** *M. vitrata* ('Mv'), *P. xylostella* ('Px'), and *S. exigua* ('Se'). Median lethal dose ($LD_{50}$) and median lethal concentration ($LC_{50}$) were determined using the hemocoelic injection and leaf-dipping methods, respectively Different letters above standard error bars indicate significant differences among means at Type I error = 0.05 (LSD test).

epoxide of the EpOMEs. A841, the mixture of propoxy derivatives at the 9th, 10th, 12th, or 13th carbon position, exhibited greater immunosuppressive and cytotoxic activities than those of 12,13-EpOME. Our previous studies showed that 12,13-EpOME was more potent than 9,10-EpOME in the immunosuppression and the cytotoxicity [17,19]. Our current study also supports the superior biological activity of 12,13-EpOME compared to 9,10-EpOME in the suppressive activity against the up-regulation of total hemocyte count in the larvae challenged by $PGE_2$. Thus, we selected two region-isomers at 12th and 13th carbons, in which PD28 with a propoxy derivative at the 12th carbon was more potent than PD23 with a propoxy derivative at the 13th carbon. Of the two possible enantiomers of PD28, the *R*-enantiomer (= AS56) was more potent than the *S*-enantiomer (= AS46). AS56 proved highly toxic to three different lepidopteran insect pests. To the best of our knowledge, no derivatives from EpOME have been developed into insecticides. A similar long-chain hydrocarbon could be juvenile hormone

**(A)**

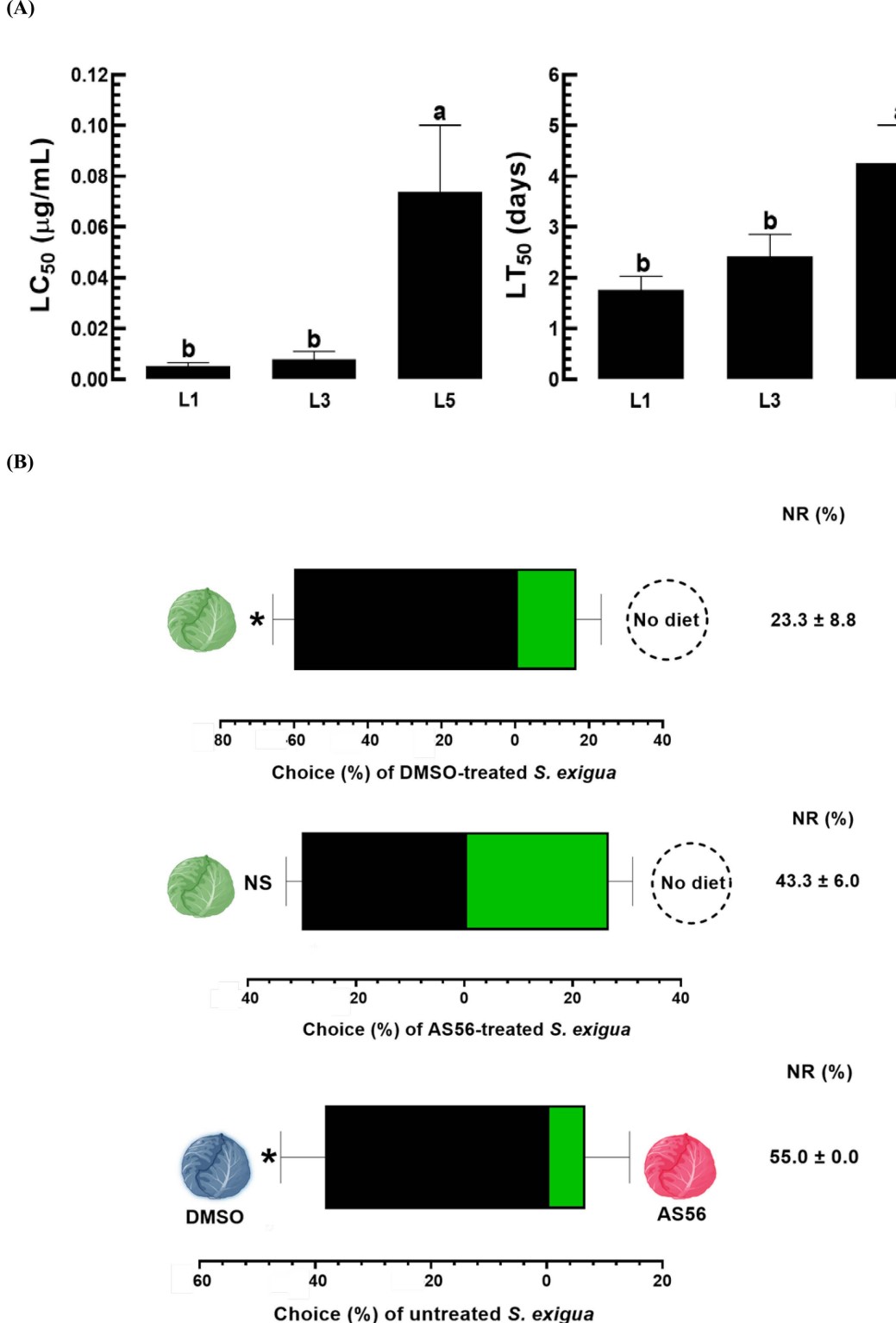

**(B)**

**(C)**

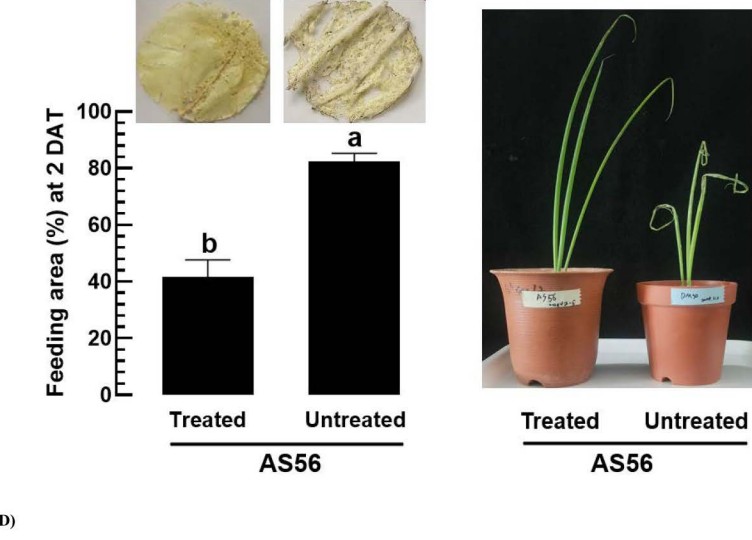

**(D)**

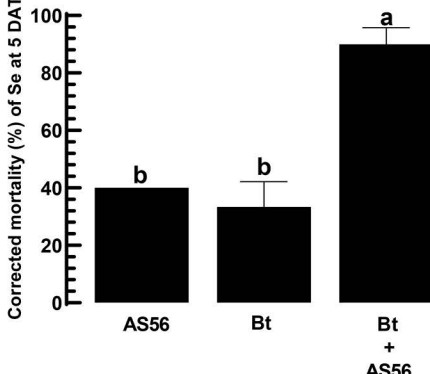

**Fig 11. Toxic activity of AS56 on the larval feeding behavior of *S. exigua*.** (A) Median lethal concentration (LC50) and median lethal time (LT50) of AS56 across various developmental stages (L1, L3, and L5) of S. exigua using the leaf-dipping method. (B) A choice test involving L3 larvae treated with either DMSO or AS56, as well as diets treated with DMSO or AS56, was conducted using a Y-tube olfactometer. Cabbages are color-coded as follows: green for untreated, red for AS56-treated, and blue for DMSO-treated. 'NR' indicates no-choice preference. (C) Decrease in feeding amount by the AS56-treated (1,000 ppm) L3 larvae on cabbage or onion. The feeding area was quantified using ImageJ software (https://imagej.nih.gov/ij). (D) Augmentation of *B. thuringiensis* (Bt, 100 ppm) virulence against L5 larvae of *S. exigua* by the addition of AS56 (10 ppm). Different letters above standard error bars or asterisks denote significant differences among means at Type I error = 0.05 (LSD test). 'NS' denotes no significant difference.

analogs such as hydroprene or methoprene [27]. However, these JH analogs are derivatives of terpenes. Furthermore, although a JH analog might induce a status-quo effect on immature development, we did not observe any JH activity in the larvae treated with AS56. Thus, we propose AS56 as a novel insecticidal compound.

Altogether, this study showed an antagonistic action of EpOMEs against $PGE_2$ signal to maintain the immune homeostasis of insects at late infection stage. Especially, the cytotoxic activity of EpOME was applied to develop a novel insecticide by modifying the labile chemical structures. Thus, an EpOME mimic, AS56, would be used for developing a novel insecticide.

## Supporting information

**S1 Table. Toxicity ($LD_{50}$ ng/larva) of EpOME alkoxides against last instar larvae of *P. xylostella* (Px), *M. vitrata* (Mv) and *S. exigua* (Se) at hemocoelic injection.**
(DOCX)

**S2 Table. Toxicity ($LC_{50}$ ppm) of EpOME alkoxides against last instar larvae of *P. xylostella* (Px), *M. vitrata* (Mv) and *S. exigua* (Se) at feeding assay.**
(DOCX)

**S3 Table. Median lethal time ($LT_{50}$) in days for last instar of *S. exigua* at 100 ppm of *B. thuringiensis* (Bt), 10 ppm of AS56 and combination of Bt (100 ppm) and AS56 (10 ppm).**
(DOCX)

**S1 Document. Chemical synthesis of EpOME alkoxides.**
(DOCX)

## Author contributions

**Conceptualization:** Niayesh Shahmohammadi, Anders Vik, Yonggyun Kim.

**Data curation:** Niayesh Shahmohammadi, Shiva Haraji, Falguni Khan, Åshild Moi Sørskår, Parastoo Ebrahimi Danielsen, Anders Vik, Yonggyun Kim.

**Formal analysis:** Niayesh Shahmohammadi, Shiva Haraji, Falguni Khan, Åshild Moi Sørskår, Parastoo Ebrahimi Danielsen, Anders Vik, Yonggyun Kim.

**Funding acquisition:** Anders Vik, Yonggyun Kim.

**Investigation:** Niayesh Shahmohammadi, Shiva Haraji, Falguni Khan, Åshild Moi Sørskår, Parastoo Ebrahimi Danielsen, Anders Vik, Yonggyun Kim.

**Methodology:** Niayesh Shahmohammadi, Shiva Haraji, Falguni Khan, Åshild Moi Sørskår, Parastoo Ebrahimi Danielsen, Anders Vik, Yonggyun Kim.

**Project administration:** Anders Vik, Yonggyun Kim.

**Resources:** Niayesh Shahmohammadi, Åshild Moi Sørskår, Parastoo Ebrahimi Danielsen, Anders Vik, Yonggyun Kim.

**Software:** Niayesh Shahmohammadi, Shiva Haraji, Falguni Khan, Åshild Moi Sørskår, Parastoo Ebrahimi Danielsen, Anders Vik, Yonggyun Kim.

**Supervision:** Anders Vik, Yonggyun Kim.

**Validation:** Niayesh Shahmohammadi, Shiva Haraji, Falguni Khan, Åshild Moi Sørskår, Parastoo Ebrahimi Danielsen, Anders Vik, Yonggyun Kim.

**Visualization:** Niayesh Shahmohammadi, Shiva Haraji, Falguni Khan, Åshild Moi Sørskår, Parastoo Ebrahimi Danielsen, Anders Vik, Yonggyun Kim.

**Writing – original draft:** Niayesh Shahmohammadi, Anders Vik, Yonggyun Kim.

**Writing – review & editing:** Yonggyun Kim.

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
