## [Decision Letter · Decision Letter 0]

9 Feb 2025

PONE-D-25-02354Antagonistic control of intracellular signals by EpOMEs in hemocytes induced by PGE2 and their chemical modification for a potent insecticidePLOS ONE

Dear Dr. Kim,

Thank you for submitting your manuscript to PLOS ONE. After careful consideration, we feel that it has merit but does not fully meet PLOS ONE’s publication criteria as it currently stands. Therefore, we invite you to submit a revised version of the manuscript that addresses the points raised during the review process.

We look forward to receiving your revised manuscript.

Kind regards,

Youming Hou

Academic Editor

PLOS ONE

Journal Requirements:

“This work was supported by a grant (No. 2022R1A2B5B03001792) from the National Research Foundation (NRF) funded by the Ministry of Science, ICT and Future Planning, Republic of Korea. This study was also funded by a research grant from Andong National University. “

5. Please expand the acronym “NRF (KR)” (as indicated in your financial disclosure) so that it states the name of your funders in full.

“This work was supported by a grant (No. 2022R1A2B5B03001792) from the National Research Foundation (NRF) funded by the Ministry of Science, ICT and Future Planning, Republic of Korea. This study was also funded by a research grant from Andong National University.”

“This work was supported by a grant (No. 2022R1A2B5B03001792) from the National Research Foundation (NRF) funded by the Ministry of Science, ICT and Future Planning, Republic of Korea. This study was also funded by a research grant from Andong National University.”

7. In the online submission form, you indicated that “The data underlying the results presented in the study are available from Yonggyun Kim.”

**Additional Editor Comments:**

Author should pay attentions to revisefd manuscript, correct the flow of paper, remove all errors and specailly take services of professional to make all corrections

Reviewers' comments:

Reviewer's Responses to Questions

**Comments to the Author**

1. Is the manuscript technically sound, and do the data support the conclusions?

Reviewer #1: No

Reviewer #2: No

2. Has the statistical analysis been performed appropriately and rigorously? 

Reviewer #1: No

Reviewer #2: No

3. Have the authors made all data underlying the findings in their manuscript fully available?

Reviewer #1: Yes

Reviewer #2: No

4. Is the manuscript presented in an intelligible fashion and written in standard English?

Reviewer #1: No

Reviewer #2: No

5. Review Comments to the Author

Reviewer #1: The abstract provides a concise summary of the study, highlighting the key findings regarding the roles of PGE2 and EpOMEs in immune regulation in Maruca vitrata. However, there are a few areas that could be improved:

The abstract is somewhat dense and could benefit from clearer language to make it more accessible to a broader audience.

The abstract mentions low nanogram levels and < 50 ppm but does not provide context for these values. It would be helpful to clarify what these concentrations mean in practical terms.

There are minor grammatical errors, such as bsract instead of abstract and findings indicate that EpOMEs are conserved which seems to be an incomplete sentence.

The introduction is well-written and provides a thorough background on insect immunity and the roles of oxylipins. However, there are a few issues:

Some information is repeated, such as the roles of PGE2 and EpOMEs, which could be streamlined for brevity.

- Incomplete Sentences: For example, These EpOMEs are findings indicate that EpOMEs are conserved is incomplete and confusing.

Some references are not properly integrated into the text, making it difficult to follow which statements are supported by which studies.

This section is detailed and generally well-organized, but there are some areas that need attention:

The description of the chemical synthesis is somewhat convoluted and could be simplified for clarity. Additionally, the reference to Fig 1 and S1 Fig is unclear without the actual figures.

The methods for quantifying cAMP and calcium ion levels are described, but the section could benefit from more detail on the statistical analysis used.

The description of the TUNEL assay is clear, but the results of this assay are not fully integrated into the results section, making it difficult to assess their significance.

The results section is comprehensive but has some issues:

Some data are presented without sufficient context or explanation. For example, the significance of the dose-dependent inhibition by EpOMEs could be more clearly explained.

The text refers to figures (e.g., Fig 10, S1-S3 Table) that are not provided, making it difficult to evaluate the results.

The statistical methods used are not clearly described, and it is unclear whether the results are statistically significant.

The discussion is thoughtful and well-reasoned, but there are a few areas for improvement:

Some conclusions seem to go beyond the data presented. For example, the potential of EpOMEs as lead compounds for novel insecticides is discussed, but the study does not provide sufficient evidence to fully support this claim.

While future research directions are suggested, they are somewhat vague. More specific recommendations would be helpful.

The discussion could better integrate the results of the TUNEL assay and other experiments to provide a more cohesive narrative.

The conclusion is generally well-written but could be improved:

The conclusion could more clearly summarize the key findings and their implications.

The suggestions for future research are somewhat generic and could be more specific.

The manuscript would benefit from clearer language and better organization, particularly in the materials and methods and results sections. The presentation of data could be improved, with more context and explanation provided for key findings. The manuscript should be revised for clarity, with particular attention to the abstract, introduction, and materials and methods sections. Author strongly recommended to take professional services of acadmicengine.com for bettler flow, grammer corrections

Reviewer #2: The manuscript contains several grammatical mistakes and awkward sentence structures. For instance, "Intracellular signaling pathway to suppress the immune responses by these oxylipins were previously unclear" should be rewritten for clarity, such as "The intracellular signaling pathway by which these oxylipins suppress immune responses was previously unclear."Ensure subject-verb agreement throughout the text (e.g., "pathway... were" should be "pathway... was").

The phrasing in several places is overly complex. Simplifying the language while maintaining scientific accuracy will improve readability.

The introduction should provide a clearer link between prostaglandin E2 (PGE2), EpOMEs, and their immunosuppressive function. Some statements feel abrupt and could benefit from smoother transitions.

The methodology regarding immune response measurements should be explained in a more structured way to ensure logical flow.

The transition from immune suppression findings to insecticidal potential should be more explicit. Currently, it feels abrupt and needs better integration into the discussion.

The manuscript does not clearly describe the figures and tables, making it difficult for readers to interpret the data. Each figure and table should be introduced with a clear rationale and explained adequately in the results section.

Statistical results should be clearly indicated in tables and figures with proper legends. Ensure consistency in formatting (e.g., p-values, standard errors, and confidence intervals should be presented uniformly).

Some details in the text about figures (e.g., hemocyte apoptosis visualization) could be better aligned with figure descriptions.

The statement that "EpOMEs suppressed the up-regulation of total hemocyte count induced by PGE2" should include statistical support and more precise wording. Does it completely suppress or partially reduce?

The claim about EpOMEs' potential as insecticidal agents requires a stronger discussion on stability, delivery methods, and environmental impact. More supporting references would enhance credibility.

The difference in immunosuppressive effects between 12,13-EpOME and 9,10-EpOME should be explained with possible mechanistic insights.

The conclusion should explicitly summarize the main findings and their significance rather than introducing new information.

Consider adding a statement on the broader implications of EpOMEs as potential biopesticides, including limitations and future research directions.

Overall, the manuscript presents interesting findings, but significant revisions are needed to improve clarity, grammar, and scientific rigor, its better to takes services of professional

6. PLOS authors have the option to publish the peer review history of their article (what does this mean? ). If published, this will include your full peer review and any attached files.

**Do you want your identity to be public for this peer review?** For information about this choice, including consent withdrawal, please see our Privacy Policy .

Reviewer #1: No

Reviewer #2: No

---

## [Author Response · Author response to Decision Letter 1]

12 Feb 2025

[Reviewer #1]

Comment #1-1: The abstract is somewhat dense and could benefit from clearer language to make it more accessible to a broader audience. The abstract mentions low nanogram levels and < 50 ppm but does not provide context for these values. It would be helpful to clarify what these concentrations mean in practical terms. There are minor grammatical errors, such as bsract instead of abstract and findings indicate that EpOMEs are conserved which seems to be an incomplete sentence.

Response:

1. The first sentence is re-written as follows: “During an infection, prostaglandin E2 (PGE2) mediates immune responses in insects and later epoxyoctadecamonoenoic acids (EpOMEs) are produced from linoleic acid to suppress excessive and unnecessary immune responses.”

2. To clarify the lethal concentration, ‘per larva’ was added to the injection dose.

Comment #1-2: The introduction is well-written and provides a thorough background on insect immunity and the roles of oxylipins. However, there are a few issues: Some information is repeated, such as the roles of PGE2 and EpOMEs, which could be streamlined for brevity. - Incomplete Sentences: For example, These EpOMEs are findings indicate that EpOMEs are conserved is incomplete and confusing.

Response:

1. To minimize redundancy, two sentences in the second paragraph is combined as follows: “Oxylipins, a family of oxygenated polyunsaturated fatty acids, include eicosanoids and epoxyoctadecenoic acids (EpOMEs), which are originated from arachidonic acid and linoleic acid, respectively [5, 6].”

2. The sentence is rephrased as follows: “These findings suggest that EpOMEs play vital roles in regulating the insect physiological processes.”

Comment #1-3: Some references are not properly integrated into the text, making it difficult to follow which statements are supported by which studies.

Response: We found the error that the reviewer indicated. The reference 18 is replaced with reference 19 in text and Reference.

Comment #1-4: This section is detailed and generally well-organized, but there are some areas that need attention: The description of the chemical synthesis is somewhat convoluted and could be simplified for clarity. Additionally, the reference to Fig 1 and S1 Fig is unclear without the actual figures.

Response: S1 Fig. is renamed as S1 Document because it is a detailed method for the compound synthesis with several NMR figures. The chemical synthetic pathway of EpOME-mimics described in the M&M is re-confirmed.

Comment #1-5: The methods for quantifying cAMP and calcium ion levels are described, but the section could benefit from more detail on the statistical analysis used.

Response: The statistical analyses are described in Results. The method is added in the M&M as follows: “cAMP and Ca2+ signals were analyzed by one-way ANOVA.”

Comment #1-6: The description of the TUNEL assay is clear, but the results of this assay are not fully integrated into the results section, making it difficult to assess their significance.

Response: The assay is added to the Results as follows: “From a TUNEL assay, the enhanced cytotoxicity of the alkoxides was attributed to their induction of hemocyte apoptosis, with AS56 being the most effective (Fig 9B).”

Comment #1-7: The results section is comprehensive but has some issues: Some data are presented without sufficient context or explanation. For example, the significance of the dose-dependent inhibition by EpOMEs could be more clearly explained. The text refers to figures (e.g., Fig 10, S1-S3 Table) that are not provided, making it difficult to evaluate the results.

Response: Fig 10 and S1-S2 Figs are separately cited in the Results to clearly describe the context. S1-S3 Tables are provided in Supplementary Information.

Comment #1-8: The statistical methods used are not clearly described, and it is unclear whether the results are statistically significant.

Response: In this comment, main issue is the statistical analysis of the continuous variables such as cAMP and calcium ion amounts. Thus we add the description as follows: “cAMP and Ca2+ signals were analyzed by one-way ANOVA.” The stat results are described in the Results.

Comment #1-9: The discussion is thoughtful and well-reasoned, but there are a few areas for improvement: Some conclusions seem to go beyond the data presented. For example, the potential of EpOMEs as lead compounds for novel insecticides is discussed, but the study does not provide sufficient evidence to fully support this claim. While future research directions are suggested, they are somewhat vague. More specific recommendations would be helpful. The discussion could better integrate the results of the TUNEL assay and other experiments to provide a more cohesive narrative.

Response: This nice comment is applied to supplement our discussion as follows: “These suggest that EpOME reduces the total hemocyte count by preventing the recruitment of the stationary hemocytes to circulatory form by the antagonistic action to the PGE2 signal. Interestingly, the majority of increased hemocytes were granulocytes in immune-challenged or PGE2-treated larvae. EpOME treatment reduced the total hemocyte count by inducing apoptosis, particularly targeting granulocytes. At 1 µg EpOME treatment, the total hemocyte count resembled that of naïve larvae, and differential hemocyte count was similarly consistent with naïve larvae. These findings suggest that EpOMEs play a pivotal role in maintaining the hemocyte populations in insects by preventing excessive recruitment and direct cytotoxic activity at late infection stage.”

Comment #1-10: The conclusion is generally well-written but could be improved: The conclusion could more clearly summarize the key findings and their implications. The suggestions for future research are somewhat generic and could be more specific.

Response: We add a conclusion based on this comment: “Altogether, this study showed an antagonistic action of EpOMEs against PGE2 signal to maintain the immune homeostasis of insects at late infection stage. Especially, the cytotoxic activity of EpOME was applied to develop a novel insecticide by modifying the labile chemical structures. Thus, an EpOME mimic, AS56, would be used for developing a novel insecticide.”

Comment #1-11: The manuscript would benefit from clearer language and better organization, particularly in the materials and methods and results sections. The presentation of data could be improved, with more context and explanation provided for key findings. The manuscript should be revised for clarity, with particular attention to the abstract, introduction, and materials and methods sections. Author strongly recommended to take professional services of acadmicengine.com for bettler flow, grammer corrections.

Response: The original manuscript was English-edited by a commercial company (Harrisco Co.). After revision based on two reviewers’ comments, the manuscript has been carefully read and confirmed by the corresponding author.

[Reviewer #2]

Comment #2-1: The manuscript contains several grammatical mistakes and awkward sentence structures. For instance, "Intracellular signaling pathway to suppress the immune responses by these oxylipins were previously unclear" should be rewritten for clarity, such as "The intracellular signaling pathway by which these oxylipins suppress immune responses was previously unclear."Ensure subject-verb agreement throughout the text (e.g., "pathway... were" should be "pathway... was").

Response: The awkward sentence is rephrased as suggested. The subject-verb agreement was confirmed through entire text.

Comment #2-2: The phrasing in several places is overly complex. Simplifying the language while maintaining scientific accuracy will improve readability.

Response: The first sentence of the abstract was rephrased to minimize the complexity.

Comment #2-3: The introduction should provide a clearer link between prostaglandin E2 (PGE2), EpOMEs, and their immunosuppressive function. Some statements feel abrupt and could benefit from smoother transitions.

Response: To make a smooth transition, two sentences in the second paragraph is combined as follows: “Oxylipins, a family of oxygenated polyunsaturated fatty acids, include eicosanoids and epoxyoctadecenoic acids (EpOMEs), which are originated from arachidonic acid and linoleic acid, respectively [5, 6].”

Comment #2-4: The methodology regarding immune response measurements should be explained in a more structured way to ensure logical flow.

Response: In M&M, immune assays are re-arranged in an order of cellular and humoral reponses.

Comment #2-5: The transition from immune suppression findings to insecticidal potential should be more explicit. Currently, it feels abrupt and needs better integration into the discussion.

Response: The results explained the biological activities of EpOMEs in an order of immunosuppression (Fig 2-5), cytotoxicity (Fig 6), and insecticidal activity (Fig 10-11). The cytotoxicity of EpOME led to the insecticidal activity.

Comment #2-6: The manuscript does not clearly describe the figures and tables, making it difficult for readers to interpret the data. Each figure and table should be introduced with a clear rationale and explained adequately in the results section.

Response: Fig 10 and S1-S2 Figs are separately cited in the Results to clearly describe the context.

Comment #2-7: Statistical results should be clearly indicated in tables and figures with proper legends. Ensure consistency in formatting (e.g., p-values, standard errors, and confidence intervals should be presented uniformly).

Response: All data related with cAMP and calcium signal are assessed by one-way ANOVA. The resulting F statistics are presented in the Results. The median lethal dose and 95% confidence interval of the insecticidal activities are described in Supplementary Information at S1 and S2 Tables.

Comment #2-8: Some details in the text about figures (e.g., hemocyte apoptosis visualization) could be better aligned with figure descriptions.

Response: Following sentence is added to Results: “Both EpOMEs induced apoptosis, as evidenced by DNA fragmentation visualized by FITC fluorescence in the TUNEL assay.”

Comment #2-9: The statement that "EpOMEs suppressed the up-regulation of total hemocyte count induced by PGE2" should include statistical support and more precise wording. Does it completely suppress or partially reduce?

Response: We showed the detailed statistical analysis as follows: “Both EpOMEs significantly inhibited the up-regulation of THC in response to PGE2, with 12,13-EpOME being more potent (F = 5.41; df = 1, 16; P = 0.002) than 9,10-EpOME (Fig 5C). Interestingly, EpOME treatment also altered the DHC (X2 = 8.5; df = 3; P = 0.037) compared to that of PGE2-induced larvae (Fig 5D). The addition of EpOMEs restored the DHC to levels similar to those of naïve larvae (X2 = 1.8; df = 3; P = 0.615).”

Comment #2-10: The claim about EpOMEs' potential as insecticidal agents requires a stronger discussion on stability, delivery methods, and environmental impact. More supporting references would enhance credibility. The difference in immunosuppressive effects between 12,13-EpOME and 9,10-EpOME should be explained with possible mechanistic insights.

Response: We add the following explanation to clarify the chemical derivatives and resulting insecticidal activities: “Our previous studies showed that 12,13-EpOME was more potent than 9,10-EpOME in the immunosuppression and the cytotoxicity [17, 19]. Our current study also supports the superior biological activity of 12,13-EpOME compared to 9,10-EpOME in the suppressive activity against the up-regulation of total hemocyte count in the larvae challenged by PGE2. Thus, we selected two region-isomers at 12th and 13th carbons, in which PD28 with a propoxy derivative at the 12th carbon was more potent than PD23 with a propoxy derivative at the 13th carbon.”

Comment #2-11: The conclusion should explicitly summarize the main findings and their significance rather than introducing new information. Consider adding a statement on the broader implications of EpOMEs as potential biopesticides, including limitations and future research directions.

Response: We add the conclusion at the end of Discussion as follows: “Altogether, this study showed an antagonistic action of EpOMEs against PGE2 signal to maintain the immune homeostasis of insects at late infection stage. Especially, the cytotoxic activity of EpOME was applied to develop a novel insecticide by modifying the labile chemical structures. Thus, an EpOME mimic, AS56, would be used for developing a novel insecticide.”

Comment #2-13: Overall, the manuscript presents interesting findings, but significant revisions are needed to improve clarity, grammar, and scientific rigor, its better to takes services of professional

Response: The original manuscript was English-edited by a commercial company (Harrisco Co.). After revision based on two reviewers’ comments, the manuscript has been carefully read and confirmed by the corresponding author.

---

## [Decision Letter · Decision Letter 1]

20 Feb 2025

Antagonistic control of intracellular signals by EpOMEs in hemocytes induced by PGE2 and their chemical modification for a potent insecticide

PONE-D-25-02354R1

Dear Dr. Kim,

We’re pleased to inform you that your manuscript has been judged scientifically suitable for publication and will be formally accepted for publication once it meets all outstanding technical requirements.

Kind regards,

Youming Hou

Academic Editor

PLOS ONE

Additional Editor Comments (optional):

I am satisfied with quality of paper

Reviewers' comments:

Reviewer's Responses to Questions

**Comments to the Author**

1. If the authors have adequately addressed your comments raised in a previous round of review and you feel that this manuscript is now acceptable for publication, you may indicate that here to bypass the “Comments to the Author” section, enter your conflict of interest statement in the “Confidential to Editor” section, and submit your "Accept" recommendation.

Reviewer #1: All comments have been addressed

Reviewer #2: All comments have been addressed

2. Is the manuscript technically sound, and do the data support the conclusions?

Reviewer #1: Yes

Reviewer #2: Yes

3. Has the statistical analysis been performed appropriately and rigorously? 

Reviewer #1: (No Response)

Reviewer #2: Yes

4. Have the authors made all data underlying the findings in their manuscript fully available?

Reviewer #1: Yes

Reviewer #2: No

5. Is the manuscript presented in an intelligible fashion and written in standard English?

Reviewer #1: Yes

Reviewer #2: Yes

6. Review Comments to the Author

Reviewer #1: manuscript issues are resolved point by point, therefore i reccmended to accpet the paper in its current form

Reviewer #2: author adress all issue which was highlighted, therfore, i reccmeded to Accept in its current form

7. PLOS authors have the option to publish the peer review history of their article (what does this mean? ). If published, this will include your full peer review and any attached files.

**Do you want your identity to be public for this peer review?** For information about this choice, including consent withdrawal, please see our Privacy Policy .

Reviewer #1: No

Reviewer #2: No

---

## [Editor Report · Acceptance letter]

PONE-D-25-02354R1

PLOS ONE

Dear Dr. Kim,

I'm pleased to inform you that your manuscript has been deemed suitable for publication in PLOS ONE. Congratulations! Your manuscript is now being handed over to our production team.

Kind regards,

on behalf of

Professor Youming Hou

Academic Editor

PLOS ONE